# Nanosensor Applications in Plant Science

**DOI:** 10.3390/bios12090675

**Published:** 2022-08-24

**Authors:** Daniel S. Shaw, Kevin C. Honeychurch

**Affiliations:** 1Department of Biology and Biochemistry, University of Bath, Bath BA2 7AY, UK; 2Faculty of Applied Sciences, University of the West of England, Frenchay Campus, Coldharbour Lane, Bristol BS16 1QY, UK

**Keywords:** nanosensors, plants, botany, nanobiotechnology, agriculture

## Abstract

Plant science is a major research topic addressing some of the most important global challenges we face today, including energy and food security. Plant science has a role in the production of staple foods and materials, as well as roles in genetics research, environmental management, and the synthesis of high-value compounds such as pharmaceuticals or raw materials for energy production. Nanosensors—selective transducers with a characteristic dimension that is nanometre in scale—have emerged as important tools for monitoring biological processes such as plant signalling pathways and metabolism in ways that are non-destructive, minimally invasive, and capable of real-time analysis. A variety of nanosensors have been used to study different biological processes; for example, optical nanosensors based on Förster resonance energy transfer (FRET) have been used to study protein interactions, cell contents, and biophysical parameters, and electrochemical nanosensors have been used to detect redox reactions in plants. Nanosensor applications in plants include nutrient determination, disease assessment, and the detection of proteins, hormones, and other biological substances. The combination of nanosensor technology and plant sciences has the potential to be a powerful alliance and could support the successful delivery of the 2030 Sustainable Development Goals. However, a lack of knowledge regarding the health effects of nanomaterials and the high costs of some of the raw materials required has lessened their commercial impact.

## 1. Introduction

Modern botany (also called plant science) is a broad and multidisciplinary topic encompassing plant biochemistry, development, chemical products, and disease. Study of the subject often uses inputs from most other areas of science and technology. Plant science has a role in the production of staple foods (e.g., wheat, oats, and rice) and materials (e.g., timber, oil, and fibre), as well as roles in genetics research, environmental management and the maintenance of biodiversity, and the synthesis of high-value compounds such as pharmaceuticals or raw materials for energy production.

Plant science has a role in addressing some of the most important global challenges we face today, including energy and food security. These global challenges can only be met in the context of a strong fundamental understanding of plant biology. This requires the comprehensive assessment of plant characteristics, including anatomical, ontogenetical, physiological, and biochemical properties—a process known as plant phenotyping [1,2]. Notably, Gregor Mendel assessed the phenotypes of pea plants to formulate the Laws of Inheritance describing equal segregation, independent assortment, and dominance of alleles [3,4]. Plant phenotyping has conventionally been performed to determine whether plant breeding programmes have resulted in an increased yield, resource efficiency gain, or enhanced desirable traits in plants such as crop species [5]. Classical plant phenotyping methods are labour intensive, costly, and time consuming [6], and so make non-destructive and real-time analysis using nanosensors an attractive proposition.

The term nanosensor is defined in this paper as a selective transducer with a characteristic dimension that is nanometre in scale. Recent advances in nanotechnology have led to the development of nanoscale sensors that have exquisite sensitivity and versatility [7]. Different types of nanosensors have been utilised in plants, including plasmonic nanosensors, Förster resonance energy transfer (FRET)-based nanosensors, carbon-based electrochemical nanosensors, nanowire nanosensors, and antibody nanosensors. Nanosensors have allowed the study of cellular functions [8] and metabolic flux [9,10], the monitoring of spatiotemporal dynamics of analytes [11,12,13], and the detection of viral and fungal pathogens [14,15,16,17,18].

A number of reviews have been published on the application of nanosensors in the plant sciences. Unlike this review, they are generally focused on the specific applications of nanotechnology, e.g., to agriculture [19,20,21] or plant pathogen detection [22,23], specific areas, e.g., *in planta* nanosensors [24], or more specific forms of nanomaterial, e.g., carbon nanotubes [25]. This review focuses on nanosensors and their applications in living plants, plant cells, plant tissues, and plant organelles. Various specific nanosensor types have been the subject of previous reviews, including FRET-based nanosensors [26,27], nanosensors based on surface-enhanced Raman spectroscopy [28], electrochemical nanosensors [29], and antibody nanosensors [30]. In this paper, the utility of nanosensor platforms for the understanding of plant cellular signalling, metabolic pathways, and phenotyping, as well as applications such as plant disease detection, are described.

## 2. The Designs and Principles of Nanosensors Used in Plant Science

Nanosensors that have been designed to interrogate plant systems promise to improve our fundamental understanding of plant biology [31]. Intracellular nanosensors are capable of detecting metabolic precursors, signalling ligands, and nutrients, and are thus capable of elucidating the complex roles of these molecules in plant systems. The emerging body of nanosensors that are used or that have a potential for use in plant science is summarised in this paper. The nanosensors, their mechanism of action, and example analytes in plants that are discussed in this paper are summarised in Table 1.

### 2.1. Förster Resonance Energy Transfer-Based Nanosensors

Optical nanosensors based on Förster resonance energy transfer have been extensively used to study protein interactions, cell contents, and biophysical parameters [32,33,34]. These sensors use light-sensitive fluorescent molecules and measure the energy transfer between them. FRET is based on the non-radiative transfer of excited state energy by dipole coupling between fluorophores if the distance between them is within a nanometre-scale range. Energy transferred from an excited donor to an acceptor molecule leads to a reduction in the donor’s fluorescence emission and an increase in the acceptor’s fluorescence emission intensities. The efficiency of energy transfer is distance dependent and can only occur over distances smaller than a critical radius known as the Förster radius—the molecular separation at which energy transfer is 50% efficient. FRET is therefore limited to short distances of up to ~10 nm for most biologically relevant fluorophores [35]. This is the same order of magnitude as the length scale of many proteins. The distance dependence and short range of FRET make for an ideal tool for studying the distance between two analytes or two sites on a specific macromolecule, such as can be found during protein conformational change [36,37], protein–protein interactions, and enzyme activity [32,38,39,40], as changes to the distance between the coupled fluorophores will be reported. Protein signalling and effector networks largely operate through conformational changes and the binding and unbinding of components, and thus FRET is an excellent tool for the study of these processes. The direct visualisation of these events is achievable using FRET with a variety of microscopy methods. Fluorophore coupling can be measured using a variety of microscopy methods and provides a sensitive and robust metric for the interaction of the biomolecules carrying the fluorescent labels.

FRET-based nanosensors can either be genetically encoded within the plant itself or added exogenously as externally synthesised compounds (Table 2). Both genetically encoded and exogenously applied FRET-based nanosensors are capable of reporting conformational changes using proteins, protein domains, nanoparticles, or molecular ligands that modulate the distance between donor and acceptor fluorescent domains of two fluorophores. Excitation of a donor fluorophore results in the transference of a fraction of the energy to the acceptor fluorophore by resonance energy transfer, which results in the fluorescence of the acceptor. Monitoring the emission peaks of the donor and acceptor enables ratiometric detection of small molecules, eliminating ambiguities in the detection by the self-calibration of two emission bands [27].

#### 2.1.1. Genetically Encoded FRET-Based Nanosensors

Genetically encoded FRET-based nanosensors are typically composed of two fluorescent proteins with spectral variations that overlap. The fluorescent proteins are generally intensiometric with one excitation (Ex) and one emission (Em) maximum. FRET-based biosensors enable a ratiometric readout when the two fluorescent proteins form a FRET pair, and the amount of energy transfer responds to an analyte. Genetically encoded FRET-based biosensors have typically used a cyan fluorescent protein (CFP) and a yellow fluorescent protein (YFP) that function as a FRET pair (as illustrated in Figure 1) with a ratiometric readout calculated from (Ex_CFP_Em_YFP_)/(Ex_CFP_Em_CFP_) [34,50]. Notably, single-fluorescent protein biosensors can also be ratiometric when the fluorescent protein has two excitation wavelengths that respond differentially to an analyte, for example, the plant-optimised pH-sensitive green florescent protein (pHGFP [51]; based on the pH-sensitive pHluorin [52]) and redox-oxidation sensitive green fluorescent proteins (roGFPs) [53,54]. A disadvantage of FRET-based nanosensors that use fluorescent proteins is the overlap in emission wavelength with chlorophyll autofluorescence (ex 410–460 nm, em 600–700 nm) and the fluorescence of cell wall components (ex 235–475 nm, em 400–500 nm) and stains. However, protocols exist to overcome this limitation [55,56]. Additionally, the implementation of FRET-based nanosensors in plants has proven difficult due to gene silencing [42,57]. The expression of FRET-based nanosensors in mutant plants deficient in gene silencing has overcome this problem and allowed the monitoring of metabolite levels in the cytosol of epidermal leaf cells and roots of plants [9,57]. Genetically encoded FRET-based nanosensors have been used to detect protein interactions with nucleic acids [41,58], to sense reactive oxygen species [59], and to monitor levels of ions or metabolites [43,60].

#### 2.1.2. Exogenously Applied FRET-Based Nanosensors

FRET-based nanosensors can be exogenously applied to plants. A variety of externally synthesised nanoparticles have been incorporated into the FRET system, including gold nanoparticles, semiconductor quantum dots (QDs), and lanthanide-doped upconversion nanoparticles, which can act as either a FRET donor or a quencher [61]. Sensors using nanoparticles overcome some of the problems of genetically encoded FRET-based nanosensors, but can also introduce others [62], such as oxidative stress [63]. Externally synthesised nanoparticles can be applied either to the roots or to the vegetative part of plants. Nanoparticles can be taken up passively through natural plant openings in the vegetative parts of plants that have nano- or micro-scale exclusion sizes, such as stomata or hydathodes [64,65]. Rhizodermis lateral root junctions may provide nanoparticle access to roots, especially near the root tip [66]. However, the presence of suberin may make the roots impermeable to nanoparticles [66]. Damaged tissues and wounds may also function as viable routes for nanoparticle internalisation in plants in both aerial and hypogeal parts [67]. Exogenously applied FRET-based nanosensors have been used for such things as the detection of plant viruses [68] and the monitoring of transgenes in plants [49,69]. In addition, nanoparticles have the potential to be used in the determination of adulterated foodstuffs. For example, upconversion nanoparticles containing Y^3+^, Yb^3+^, and Er^3+^ have been applied utilising p-toluidine as a recognition molecule for the determination of furfural [70]; a product of the dehydration of sugars. Furfural is an aromatic aldehyde formed by the pyrolysis of organic matter, and presents both toxic and carcinogenic properties. An absorption peak appeared when furfural interacted with p-toluidine, leading to the quenching of the emission at 539 nm, allowing for the determination of furfural. The nanosensor reportedly showed potential in the determination of furfural adulterated foodstuffs, such as cookies, honey, and fruit wine.

### 2.2. Surface-Enhanced Raman Scattering Nanosensors

Surface-enhanced Raman scattering (SERS) is a sensitive non-destructive spectroscopic technique able to detect analytes at the single-molecule level [28]. Raman scattering is the inelastic scattering of photons by matter when illuminated. In this process, there is an exchange of energy and a change in the light’s direction. The energy difference between the scattered light and the incident light is due to the interaction of photons with the vibrational states of matter. This effect involves vibrational energy being gained by a molecule as incident photons are shifted to lower energy state. The effect was theoretically postulated by A. Smekal in 1923 [71], and experimentally demonstrated by C. V. Raman in 1928 [72]. The effect can be exploited to gain information about materials by performing one of the various forms of Raman spectroscopy, such as SERS spectroscopy. However, it can be challenging to obtain spectra from analytes in low concentrations using Raman scattering as the signals can be weak.

Surface-enhanced Raman scattering spectroscopy is based on the amplification of the Raman signal for molecules adsorbed on a nanostructured metallic surface. Raman signals of molecules adsorbed on the surface of metal nanoparticles can be enhanced 10^14^−10^15^-fold; sufficient sensitivity to detect single molecules [28,73,74]. The SERS effect is proposed to be the result of two enhancement mechanisms: an electromagnetic mechanism and a chemical mechanism [75,76]. The electromagnetic mechanism is generally recognised to be essential for SERS, whilst the chemical mechanism plays a role in the ultrasensitive application of the technique [77,78,79,80,81]. The electromagnetic mechanism of SERS enhancement results from the redistribution of the electromagnetic field around metallic nanostructures (also known as optical enhancers) in the 10–200 nm range. This effect is mediated through the resonance of the incident light with the surface plasmon resonances of the metal. The chemical mechanism results from the interaction between the metal surface and molecules adsorbed to it. This accounts for deviations in the relative intensities (and frequencies) in the vibrational modes of a molecule when compared with normal Raman spectra. In addition, the chemical mechanism of SERS enhancement is used to explain discrepancies between the maximum enhancement factors obtained theoretically from the electromagnetic calculations and those found experimentally [73,74]. The chemical mechanism is associated with two processes: charge-transfer processes and the formation of analyte–surface complexes [82,83]. The charge-transfer states involve transitions from the Fermi level of the metal to an unoccupied orbital of the molecule (or vice versa). The formation of a surface complexes between the metal and the analyte molecule results in a change in the properties of the molecule (such as the possibility of resonance Raman scattering). In plants, most peaks of the SERS spectra are attributed to adenine-containing materials, flavins, chlorophyll, and lipids [84]. SERS has been used extensively for the detection and determination of a wide range of biological molecules in plants [85], such as for plant hormones [86,87], and have practical applications in food-safety diagnostics such as for pesticide detection [88,89] (Table 3).

### 2.3. Electrochemical Nanosensors

Electrochemical nanosensors typically consist of a working electrode, counter electrode, and reference electrode. Amperometric and voltammetric techniques have been shown to be useful tools in the qualitative study of plant sensing [90]. These techniques are also useful in other fields where industrial environmental sensors are used (e.g., air and water). Electrodes designed with nanomaterials have a relatively high active surface area conferring higher sensitivity. Recent advances with electrochemical nanosensors are mostly attributed to advances in the research of metallic nanoparticles such as gold [91,92,93] and nanocarbon materials, including carbon nanotubes and graphene-based materials, due to their unique electronic properties [94,95].

Electrochemical detection is an attractive method for the detection of biological molecules in plants due to its high sensitivity and the capacity for direct data analysis [30]. Electrochemical nanosensors are capable of indicating plant growth and environment conditions by detecting a range of biological molecules, such as the plant hormones ethylene [96,97] and auxin family member indole-3-acetic acid [98,99], enzymes such as urease which is involved in plant metabolism of urea [100,101,102], and other biological molecules such as vitamin C [103,104,105], citric acid [106], and glucose [94,107,108,109] (Table 4). Moreover, electrochemical nanosensors can follow reactive oxygen species (ROS) and ROS-related products in plants such as hydrogen peroxide (H_2_O_2_) [99,110,111], oxygen (O_2_) [112,113,114], plant thiols [115,116], and glutathione [117], which can indicate a plant’s metabolomic, stress, and environmental conditions. In addition, electrochemical nanosensors can directly analyse the environment in which plants find themselves. For example, electrochemical nanosensors can determine soil contents, which is important for obtaining the optimal crop production rates. Many plants experience oxidative stress upon exposure to heavy metals that can lead to cellular damage [118]. Electrochemical nanosensors can determine the concentration of heavy metal ions in the soil [119,120]. Moreover, determinable soil contents include nutrient ions, such as H^+^, K^+^, and Na^+^, in the soil of the plant [121], which are required for optimal growth.

Chemiresistive sensors are another type of electrochemical nanosensor. These sensors monitor the change in electrical resistance between two electrodes caused by the adsorption of target molecules to the sensing material. The sensing materials could be semiconductors, such as carbon nanotubes, as well as conducting polymers, such as polypyrrole. This technique is useful for gas sensing due to its high sensitivity—down to the parts-per-trillion (10^−12^) range. Chemiresistive sensors have been used to sense the gaseous plant hormone ethylene [122,123,124,125], as well as other volatile organic compounds that are produced by plants [126,127].

To study plants with electrochemical nanosensors, plant tissues are commonly homogenised in a known electrolyte solution [128]. It is also possible to study plants indirectly with electrochemical nanosensors by focussing on molecules used in plant growth and development found in the soil [129]. The development of nanosensors designed to reside within plants has allowed the study of plants in situ without the need to homogenise plant tissues in electrolyte solution for further analysis. One such sensor based on photoactive nanomaterials (molybdenum-doped bismuth vanadate; BiVO_4_) acted as a sensing unit in a photoelectrochemical platform for antioxidant capacity evaluation in fruit [130].

### 2.4. Piezoelectric Nanosensors

Piezoelectricity is a coupling of mechanical and electrical behaviours of materials and refers to a reversible process in which an electric charge accumulates in certain solid materials in response to applied mechanical stress, and vice versa, the contraction or elongation of a solid material when positioned in an electric field [131]. The piezoelectric effect can be exploited for many applications, such as for sensors. A piezoelectric sensor is a device that uses the piezoelectric effect to measure changes in pressure, acceleration, temperature, strain, or force by converting them to an electrical charge.

A nanometre-sized force/pressure piezoelectric sensor capable of measuring forces in the nanonewton range and even smaller has been demonstrated [132]. Piezoelectric nanosensors are of nanometre scale, but it should be noted that the detection process typically goes through a charge voltage amplifier which is of centimetre scale. Piezoelectric nanosensors have been used to measure the biomechanics of plants such as the Venus flytrap (*Dionaea muscipula*; [133,134]). The knowledge gained from studying biomechanics, morphing structures, mechanosensors, and osmotic motors in plants is a useful input for designing adaptive structures and intelligent materials. They have also been used to detect staphylococcal enterotoxin B in apple juice [135], and for phytopathology [136].

### 2.5. Nanoparticles in a Living Plant or Plant Organelles

The introduction of nanoparticles into desired plant tissues is necessary for the effective use of nanosensors to monitor plant biochemical pathways and organic compounds. The plant cell wall is a barrier absent in many other organisms and can impede the internalisation of nanoparticles by plant cells and is also able to mediate nanoparticle effects on the plant. The localisation of nanoparticles within plants, plant cells, and plant organelles is also important for the effective spatiotemporal sensing of plant analytes. The trafficking and localisation of nanoparticles has been studied in the *Cucurbitaceae* (*Cucurbita maxima* and *Cucurbita pepo*) [137,138]. The behaviour of nanomaterials within plant organelles such as the chloroplast has also been studied [139,140]. Particle parameters such as the size and the magnitude can determine whether a particle is spontaneously and kinetically trapped within plant organelles [140]. The effects and toxicity of nanoparticles on plant growth and functions continues to be an active area of research [141]. The toxicological effects of nanoparticles depend on the particle size, shape, surface area, chemistry, and the surface functionalisation of the nanoparticles [142,143,144]. In addition, impurities introduced during the synthesis of the nanoparticles can also result in toxicological effects, which has led to erroneous generalisations about the toxicity of nanoparticles [141].

## 3. Nanosensor Applications in Plants

The detection of analytes is optically difficult *in planta* due to tissue thickness and the presence of photosynthetic pigments in plant tissues. Nanosensors are well-suited for the detection of analytes as they are easily embedded in plant tissues. They are thus well-suited for in vivo studies of cellular signalling and metabolism. Examples of the nanosensor applications in plants are highlighted in the following subsections.

### 3.1. Detection of Molecular Oxygen

Molecular oxygen is the terminal electron acceptor in the electron transport chain during aerobic respiration. Plants (as well as algae and photosynthetic bacteria) are able to produce oxygen via photolysis, which is part of the light-dependent reaction of photosynthesis. However, the availability of oxygen in the atmosphere is still an essential substrate for plant metabolism as photosynthetic activity varies in tissue and there are times when plants are not photosynthetically active [145]. Many stresses, for example flooding, can also result in hypoxia in plant tissues. In addition, differential patterns of the abundance of oxygen occur in organs and meristems and the regulation of oxygen status is mechanistically related to plant development [146].

Extensive work on oxygen sensing has utilised Clark-type polarographic electrode sensors to detect a current flow caused by the chemical reduction of oxygen to water [147]. These microelectrodes have been used to determine the rates of photosynthesis and respiration by potato leaf protoplasts [148], measure the respiration rate of mitochondria extracted from pea shoots [149] and the leaves of *Arabidopsis thaliana* [150], as well as to measure alternative oxidase activity in soybean cotyledons and roots [151]. However, these electrodes have practical limitations when compared to optical sensors; they are invasive and can require extensive sample preparation, and they consume oxygen causing experimental errors when used in a living cell. Some of these problems can be overcome by using nanosensors. At present, two categories of nanosensor are being utilised to assess oxygen distribution inside tissues, namely electrochemical and optical systems.

Electrochemical nanosensors (carbon-filled quartz micropipettes with platinum-coated tips) have been used to detect a considerable drop in oxygen concentration at the surface of *Chara corallina* internodes in response to micro-perforation of the cell wall [114]. The decline in oxygen concentration at the wounding site could be due to several causes, such as the stimulation of the plasma membrane NADPH oxidase, and modulation of antioxidant systems.

Optical nanosensors for O_2_ also have features that make them an attractive alternative to the Clark-type polarographic electrode sensors, whilst enabling oxygen to be sensed on a nanoscale and to be imaged over large areas. Probes encapsulated by biologically localised embedding (PEBBLEs) are a prominent class of this type of nanosensor. The sensing elements, i.e., fluorescent dyes of PEBBLEs are encapsulated within an inert matrix which reduces dye leakage [152]. In addition, the protective shell retains stability and prevents interference with other proteins [153]. However, PEBBLEs commonly emitted a red phosphorescence signal that interfered with the autofluorescence of the plant chlorophyll when applied in plant cells. To circumvent the interference of the plant autofluorescence, a microbead-based probe was developed for application in plant- and algae-based systems that utilised the two-frequency phase modulation technique [154]. Nanoparticle oxygen sensors typically have excellent brightness and photostability, and are relatively simple to produce, with the added benefit that long-term storage is possible [145]. However, the size of the probes—ranging from 20 to 600 nm in diameter [152]—may lead to cell damage, and hence limit their application in living plants [154].

### 3.2. Water and Humidity Nanosensors

Utilising an aluminium oxide nano-porous ceramic plate (mean pore size: 30 ± 15 nm), an optical-based sensor for the direct, continuous monitoring of soil water has been reported [155]. The nano-porous ceramic disc was used in conjunction with a silicon diaphragm and a miniature optical displacement detection unit, composed of an integrated light source and photodetector. When the sensor is buried into unsaturated soil water, a negative pressure inside the reservoir is established, inducing diaphragm bending. The resulting displacement caused by the diaphragm could then be used to measure the dry soil saturation.

Subsequently, Leone et al. [156] presented a compact innovative optical, low-cost platform for soil water content measurement based on a nano-porous ceramic disc, however, in this case, in connection with an engineered optical fibre with near-infrared-based detection. The sensor consisted of a Y-shaped bifurcated cable housing two fibres in a single body. These fibres were placed side-by-side with one connected to the light source and the other to the detector. The common leg allowed for illumination of the fibres and served to collect the light from the disc. For a soil testing, the sensor was placed in a protective PVC tube buried in a soil tank.

Lan et al. [157] have reported on the fabrication of a capacitive wearable graphene-based plant humidity sensor. The capacitive-type humidity nanosensor was fabricated using laser direct writing technology on a polyimide film to give a graphene interdigital electrode (LIG-IDE). An aqueous solution of graphene oxide (GO) was then drop-cast onto the surface of LIG-IDE to act as the sensing element of the humidity sensor. The flexible GO humidity nanosensor could be readily attached to the surface of plant leaves without reportedly adversely affecting the growth of the plant (Figure 2). The nanosensor could be combined with wireless devices to give an integrated system.

### 3.3. Detection of Adenosine Triphosphate

Adenosine triphosphate (ATP) is an organic chemical that provides energy to drive many processes in living cells. ATP is dephosphorylated either to adenosine diphosphate or to adenosine monophosphate when consumed in metabolic processes. In plants, ATP is synthesised in chloroplasts and mitochondria. A decrease in cytoplasmic ATP levels following the addition of oligomycin A (a mitochondrial ATP synthase inhibitor) was detected using a chimera of enhanced *Renilla* luciferase and a fluorescent protein with high bioluminescence resonance energy transfer efficiency (Venus) [44]—an optical nanosensor based on FRET. In addition, ATP production in chloroplasts during photosynthesis has been visualised in transgenic Arabidopsis plants by targeting this optical nanosensor to the chloroplast stroma by using a transit peptide fusion [44].

### 3.4. Detection of Calcium Ions

Calcium plays an important role in signal transduction pathways. Calcium ions are involved in multiple plant processes such as stomatal closing, cellular division, and cell signalling. For example, in the process of stomatal closing, free Ca^2+^ ions enter the cytosol from both outside the cell and internal stores following abscisic acid signals to the guard cells. This has the effect of reversing the concentration gradient and K^+^ ions begin exiting the cell. The loss of solutes makes the cell flaccid and closes the stomatal pores. FRET-based genetically encoded sensors allow high-resolution live cell imaging of Ca^2+^ dynamics [45]. Analysis of Ca^2+^ dynamics in *Lotus japonicus* revealed distinct Nod factor-induced Ca^2+^ spiking patterns in the nucleus and the cytosol.

### 3.5. Detection of Reactive Oxygen Species

Nanosensors are also capable of detecting reactive oxygen species. Superoxide anion (O^2•−^), singlet oxygen (^1^O_2_), hydroxyl radical (•OH), and hydrogen peroxide (H_2_O_2_) are the major ROS in plants. ROS can be produced during normal cellular metabolism at cellular sites such as the chloroplast, mitochondria, peroxisomes, and apoplast [158,159,160]. ROS are involved in numerous signalling pathways in plants, including those involved in plant development, cell death, and responses to various types of stress [161,162]. ROS and redox potentials can be measured using genetically encoded ratiometric single-fluorescent protein sensors, such as roGFPs [53,54], to monitor the glutathione redox state [163,164] and extrinsic sensors, such as HyPer—a single fluorescent ROS sensor that directly reports H_2_O_2_. In plants, roGFPs have been extensively used to determine glutathione redox potential [164,165,166,167]. HyPer has been used to detect H_2_O_2_ changes in Arabidopsis guard cells and roots [168,169,170]. In plants, oxidative bursts can play significant roles in plant disease defences and signal transduction. The real-time monitoring of oxidative burst from single plant protoplasts has been achieved using electrochemical sensors modified with platinum nanoparticles [111].

### 3.6. Detection of Nitric Oxide

Nitric oxide is a gaseous reactive nitrogen species that acts as a signalling molecule throughout the plant life cycle. Nitric oxide is involved in a range of physiological activities in plants, ranging from seed germination to senescence and programmed cell death [171]. Furthermore, nitric oxide also acts as a signal in response to biotic and abiotic stresses [172]. However, the precise role of nitric oxide in signalling pathways and ideal methods of measurement remain an active area of research. One method of measurement employs semiconducting single-walled carbon nanotubes as signal transducers for nanosensors. Semiconducting single-walled carbon nanotubes have been exploited for near-infrared fluorescence monitoring of nitric oxide in *A. thaliana* [139]. This technique employed corona phase molecular recognition, which uses the specific adsorption of a compositionally designed polymer at a nanoparticle interface to enable recognition.

### 3.7. Detection of Plant Hormones

It is possible to detect a range of plant hormones, such as strigolactones [11], ethylene [96,97], and auxin [98,99], using a variety of nanosensors. Plant hormones (also known as phytohormones) are signal molecules produced within plants and are involved throughout a plant’s growth and development from embryogenesis [173] to reproductive development [174], as well as in biotic [175,176] and abiotic stress tolerance [177,178]. Nanosensors offer opportunities to study plant hormones and signalling mechanisms in vivo.

Strigolactones are a group of chemical compounds produced by a plant’s roots [179] and represent a class of plant hormones that regulate developmental processes and play a role in the response of plants to various biotic and abiotic stresses [180]. They have been identified as being involved in three different processes: the promotion of the germination of parasitic organisms that grow in the host plant’s roots [179,181,182], in the recognition of the plant by symbiotic fungi [179,183], and the inhibition of plant shoot branching [179,184,185]. One such parasitic organism that grows in the host plant’s roots is Striga [179,181,182]. The use of an optical nanosensor (a fluorescence turn-on probe called Yoshimulactone Green) allowed for the spatiotemporal monitoring of strigolactone levels in germinating Striga seeds [11]. The recognition of Yoshimulactone Green by strigolactone receptors and its subsequent hydrolysis generates detectable fluorescent products. In addition to the spatiotemporal monitoring of strigolactone levels, Yoshimulactone Green was used to determine specific strigolactone receptors [11].

Ethylene regulates many aspects of the plant life cycle, including seed germination, root initiation, flower development, fruit ripening, senescence, and responses to biotic and abiotic stresses [186]. Ethylene is widely used in agriculture to force the ripening of fruits [187]. Chemiresistive sensors have been used to sense the gaseous plant hormone ethylene [122,123,124,125] and have shown a reliable ethylene response toward different fruit types such as banana, avocado, apple, pear, and orange.

Auxins are plant hormones that influence multiple aspects of plant development such as cell enlargement, bud formation, and root initiation. A sensor employing platinum black and carbon nanotube surface modifications characterised auxin flux in 3- to 5-day roots non-invasively [98]. Moreover, a sensor utilising a porous graphene bionanocomposite of porous graphene, gold nanoparticles, and anti-indole-3-acetic acid antibody for sensitive and label-free amperometric immunoassay of indole-3-acetic acid (IAA; an auxin-class hormone) was reported to have a low detection limit and can been applied to the detection of IAA in plant sample extracts [188].

Gibberellins are plant hormones that promote organ growth and regulate a variety of developmental processes. Mutants defective in GA biosynthesis are characterized by reduced elongation of roots, stems, and floral organs [189]. A FRET-based nanosensor has been developed for the high-resolution quantification of spatiotemporal gibberellin distribution [46]. To develop the FRET-based nanosensor for gibberellin, plant hormone receptors were used as sensory domains. The GIBBERELLIN INSENSITIVE DWARF 1 (GID1) protein is a soluble receptor protein that interacts with gibberellins in an internal binding pocket [190]. Gibberellin binding promotes GID1 interaction with members of the DELLA family of growth regulators in plants. The GID1–gibberellin complex leads to the degradation of the DELLA protein after binding to the N-termini of the DELLA protein [191]. The Arabidopsis gibberellin perception machinery was adapted into a conformationally dynamic gibberellins binding domain within a FRET nanosensor by fusing GID1 variants to DELLA N-termini. This fusion converts the gibberellin-dependent intermolecular interactions into gibberellin-dependent intramolecular structural rearrangements. In this way, the nanosensor responds to nanomolar concentrations of bioactive gibberellins with an increase in the emission ratio and has been used to report gibberellin distribution and gradients in vivo in multiple tissues [46].

Salicylic acid (SA) is an important plant hormone that is best known for mediating host responses upon pathogen infection [192]. Derivatives of salicylic acid can be found in food products, medicines, cosmetics, and preservatives. A structure-switching aptamer-based nanopore thin film sensor has been developed for the detection of salicylic acid in plant extracts [193]. Due to its small size and scarcity of reactive groups for immobilization, salicylic acid is reportedly a challenging target for aptamer selection using conventional systemic evolution of ligands. However, the authors Chen et al. reported the development of a nanopore thin film sensor platform capable of determining levels as low as 0.1 μM salicylic acid, which showed good selectivity towards salicylic acid and its metabolites. It was shown possible to determine salicylic acid in Arabidopsis and rice using only about 1 μL plant extracts, with an assay time of less than 30 min.

### 3.8. Determination of Fruit Ripening

The perishability of fruits is a long-standing supply chain issue, causing a sizeable proportion of harvested fruits to be discarded before distribution to consumers [194]. As discussed above, ethylene is a major plant hormone that dictates fruit ripening [186]. It is possible to regulate the ripening dynamics of climacteric fruits through the manipulation of ethylene concentration—a technique widely used to extend shelf-life and ensure shelf-maturity. The mechanisms of fruit ripening and spoilage have been well studied [187,195,196,197]. Ethylene concentrations at 1 parts-per-million (ppm; 10^−6^) have been shown to initiate the ripening of climacteric fruits [197], while ethylene-sensitive fruits such as bananas and kiwis were found to be affected by sustained exposure to 10 parts-per-billion (ppb; 10^−9^) ethylene [198,199]. For this reason, ethylene concentration has been used to identify an optimal harvest period [200], define ideal storage conditions [201,202], and control the speed of ripening. Chemiresistive sensors have demonstrated ethylene detection as low as 0.5 ppm [122] as well as their utility in the determination of fruit ripeness [122,123,124,125].

Equally, the deliberate or accidental adulteration of plant oils can have notable effects on the supply chain. Spaniolas et al. [203] have developed lab-on-a-chip based technology for the determination of the adulteration of plant oils. The methodology was based on the combinatorial use of a polymerase chain reaction (PCR) assay with a capillary electrophoresis lab-on-a-chip based assay. The variability in the length of chloroplast *trn*L intron among different plant species was used for the authentication of oils. The application of the assay on DNA extracted from different plant-derived oils was undertaken and determined to be capable of detecting the adulteration of olive oil with various other plant oils.

### 3.9. Plant Pathogen Detection

Crop losses to plant pathogens represent a significant cost to farmers. Nanosensors offer the opportunity to detect pathogens so that containment is possible. Diagnosis is currently performed using microbiological or PCR-based techniques [204,205,206,207]. While these techniques are often sophisticated and accurate, they can also be time-consuming. Nanosensors offer an alternative method of detection as they allow for the rapid detection of fungi, bacteria, and viruses in plants [14,68].

Sensors encompassing fluorescent silica nanoparticles combined with antibody molecules have been used to detect *Xanthomonas axonopodis* pv. *vesicatoria*, which causes bacterial spot disease in Solanaceae plants [15]. In addition, a nanosensor based on fluorescently labelled-DNA oligonucleotide conjugated to 2-nm gold nanoparticles detected phytoplasma associated with the Flavescence dorée disease of grapevine [17]. Moreover, an electrochemical sensor utilising gold nanoparticles was shown to be capable of detecting *Pseudomonas syringae* in *A. thaliana* by differential pulse voltammetry [16]. Nanosensors are also available for mycotoxin detection. The 4mycosensor is a competitive antibody-based assay capable of detecting ZEA, T-2/HT-2, DON, and FB1/FB2 mycotoxin residues in corn, wheat, oat, and barley [18,208]. QD-based biosensors have been used to detect *Cowpea mosaic virus* [209], *Cauliflower mosaic virus* [210], *Citrus tristeza virus* [47,211,212], *Grapevine virus A* [48], *Tomato ringspot virus* [213], *Bean pod mottle virus* [213], and *Arabis mosaic virus* [213].

The synthesis of gold nanoparticle glycoconjugates based on functionalised sugars was recently reported [214]. The gold nanoparticle glycoconjugates were subsequently employed in the development of a sensor for the detection of the spores and hyphae of the blue-green mould *Penicillium italicum* in fruit [215]. This was based on the recognition of lectin. Lateral tests using standalone poly(amic) acid (PAA) membranes on glass and 96-well polystyrene plates utilising paper electrodes were investigated. Both substrates were functionalised with derivatised sugar-based ligands and stained with gold nanoparticles. The authors reported strong signals for 104 spores/mL of *P. italicum* isolated from infected lemons. The 96-well plate approach was found to be the most sensitive approach with a detection limit of 4 × 10^2^ spores/mL, with a linear range from 2.9 × 10^3^ to 6.02 × 10^4^ spores/mL. A standard deviation of less than 5% for five replicate measurements was reported. The fungi *P. italicum* was successfully identified over related fungi species *Trichaptum biforme*, *Glomerulla cingulata* (*Colletotrichum gloeosporioides*), and *Aspergillus nidulans*. The authors concluded that this specificity resulted from the sugar ligands employed in the synthesis of the gold nanoparticles and was unaffected by their size and shapes [216].

The pathogen, *Xylella fastidiosa* subsp. *pauca* strain CoDiRO, is responsible for olive quick decline syndrome (OQDS). This represents a great threat to agricultural-based economies such as that of South Italy. The bacteria can also infect other plant species. As a result, quarantine programs have been put in place in parts of Italy. Symptoms of OQDS include leaf scorching and wilting of the canopy, and can appear months after the initial infection with some hosts also being asymptomatic. Consequently, sensors for the rapid and early screening of plants are highly desirable. Determination of *X. fastidiosa* is normally undertaken by ELISA and PCR. Chiriacò et al. [217] have compared these two standard methods with a lab-on-a-chip assay for the determination of *X. fastidiosa* detection in leaf samples. The developed lab-on-a-chip includes a microfluidic module, and its performance is competitive with conventional diagnostic methods in terms of reliability, but with further advantages of portability, low costs, and ease of use. Thus, the proposed technology has the potential to be a useful assay method for large-scale monitoring programs.

The lab-on-a-chip system used for *X. fastidiosa* detection was based on a polydimethylsiloxane (PDMS) microfluidic module with microchannels and 20 μL microchambers fabricated by replica moulding. A system of inlet and outlet holes was incorporated to allow for the delivery of test samples directly on the surface of an interdigitated metallic microelectrode array, fabricated via optical lithography on a glass substrate. The device layout has a central inlet aperture and four peripheral outlets per side, allowing for the contemporaneous testing of different samples (Figure 3). The central inlet was used to perform functionalisation steps and to insert the sample to be measured and delivered to the four chambers, allowing for measurements to be made either in quadruplicate or for separate samples. The interdigitated electrodes were further functionalised with *X. fastidiosa* specific antibodies. Quantification was obtained by impedance spectroscopy, following the addition of a 1:1 solution of hexacyanoferrate (II/III).

The application of microfluidic chip for the high-throughput phenotyping of *A. thaliana* [218]—a commonly used as a model organism in plant biology and genetics—has been reported. Multiple Arabidopsis seeds were germinated and propagated hydroponically in the chip, making it possible to continuously investigate phenotypic changes in plants at the whole organismal level and at the cellular level. Reportedly, the Arabidopsis plants grown in the device maintained normal morphological and physiological behaviour, and phenotypic variations between wildtype and mutant plants were measurable. The timeline for the plant’s different developmental stages in the chip was reported as being highly comparable to growth recorded on a conventional agar plate. Using the microfluidic device, it was shown possible to identify changes occurring during plant–pathogen interactions. The authors postulate their prototype plant chip technology could be used for the basis of a high-throughput and precise plant phenotyping device.

Julich et al. [219] have developed a lab-on-a-chip approach for the rapid nucleic acid-based diagnosis *Phytophthora*—a genus of plant-damaging oomycetes. PCR and hybridisation steps were performed consecutively within a single chip consisting of two layers; an inflexible and a flexible one, with integrated microchannels and microchambers containing a polymeric component, with integrated half channels placed on the inflexible component containing the DNA microarray. The 32 measurement points on the chip allow the incorporation of five different capture sequences in quadruplicates plus negative and positive controls and untreated electrode gaps to monitor the background signal. This allows for at least five different DNA fragments to be tested in parallel on the chip in the current setup. Data from the microarray was collected electrochemically, based on the deposition of elementary silver by enzymatical catalysation. After an initial 5-min period of silver deposition, increased conductivity values were recorded at the positive control. After a period of 8–10 min of total silver deposition, conductivities of 10^−4^ to 10^−2^ Siemens were reported only for fully complementary capture sequences. Incomplete complementary sequences and negative controls showed no increase in conductivity within 10 min at all measurement points. The electrical readout was reported to be simpler and faster than PCR technology generally used for such investigations. Deposited silver spots were reported to show long term stability compared to fluorescent signals that are affected by bleaching. The specificity of the lab-on-a-chip system was investigated for the determination of five species of *Phytophthora*. However, two of these species were reported to give signals below the threshold.

### 3.10. Fertiliser and Pesticide Management

The application of fertilisers plays an important role in increasing agricultural production. However, excessive use of fertilisers can alter the chemical ecology of soil and reduce the amount of land available for crop production [220]. Non-destructive nanosensors capable of transducing plant signals into digital signals permit the establishment of direct communication between plants and growers, facilitating controlled fertiliser release while minimising their use. In addition, electrochemical nanosensors can determine the concentration of various ions in the soil and so can be used to inform on appropriate levels of fertiliser applications. Ion-selective electrodes have been used to monitor the sap of potatoes [221,222] and broccoli [223]. Electrochemical nanosensors can detect heavy metal ions [119,120], as well as ions used for plant growth, such as H^+^, K^+^, and Na^+^ [121,221]. It is possible to incorporate these ion-selective electrodes into greenhouse industry systems to manage liquid fertilisation strategies [121,129,224].

Pesticides are widely used in modern agriculture. The adverse effects of pesticides on the agricultural ecosystem have been a matter of concern in recent decades, and have established the need for monitoring programmes to determine the fate and accumulation of pesticides in the soil [225]. Understanding the behaviour of pesticide translocation is significant for effectively applying pesticides and reducing pesticide overexposure. SERS utilising gold nanoparticles has been used in the real-time monitoring of pesticide translocation in tomato plant tissues, including in the leaves and flowers [226]. In addition, flame aerosol technology has been used to rapidly self-assemble uniform SERS sensing films to detect pesticides [89]. This technology combines particle synthesis and facile film fabrication in a cost-effective and single process step. To synthesise nanoparticles, solution containing Ag and Si precursors was fed through a capillary, atomised using pure oxygen into fine droplets, and ignited. The nanoparticles were generated through droplet evaporation and combustion, particle nucleation, growth by coalescence and sintering, aggregation, and agglomeration [227]. At the same time, the films are generated by the depositions of nanoparticles on a temperature-controlled glass substrate by thermophoresis to produce highly uniform and reproducible SERS sensing surfaces. Pesticide residues collected from the surface of an apple and dissolved in an ethanol solution were applied to the SERS substrate for SERS measurements to be taken. The presence of the pesticide parathion-ethyl was verified, demonstrating an application in food-safety diagnostics for pesticide detection on fruit surfaces [89].

Insect pheromones are used in pest management programs, typically for pest detection and monitoring, and deciding the timing of pesticide spray programs. Recently, a cantilever-based gas nanosensor coated with a polyaniline and sodium polystyrene sulfonate nanocomposite and a polyaniline-silver nanohybrid was reported for the monitoring of a pheromone released by the neotropical brown stink bug, *Euschistus heros* (F.) [228]. Rubber septa insect pheromone dispensers were impregnated with 2,6,10-methyl trimethyltridecanoate, which is the main component of the sexual pheromone of *E. heros*. Over a period of two months, the cantilever nanosensors showed a daily reduction in resonance frequency when exposed to the pheromone, which was not observed in the control cantilever. The authors reported that relative humidity did not influence the nanosensors resonance frequency, and the cantilever nanosensors were stable for twelve months.

### 3.11. Future Directions

There is a tremendous range of activity and exciting research in the field of nanosensors and plant science. Research has begun to address important issues relating to the development, production, and application of nanosensors. This has highlighted the need for co-operation of researchers from across fields and the formation of multidisciplinary teams to advance the development of both nanotechnology and plant science.

Nanosensors promise to deliver precision measurements to optimise the growth and productivity of plants in agriculture, forestry, and research fields. Stakeholders, e.g., farmers and scientists, are ready to embrace these novel analytical tools to guide their management decisions, but few examples of nano-based plant sensors have reached the market. There remain challenges to the widespread, real-world applications, primarily related to integration of nano-sensing elements into analytical devices and fabrication on an industrial scale. The lack of knowledge of the health effects of nanomaterials and the high costs of some of the raw materials has adversely affected the commercial impact of nanosensors. This has been exacerbated by the associated high manufacturing and scale-up costs of nanomaterials in a low-margin product sector. In addition, the lack of defined markets (common to any new technology) needed to make plant nanosensors attractive to investors and manufacturers could be contributing to the slow pace at which nanosensors are being brought from the proof-of-concept stage to full deployment in the field. However, nanosensors continue to attract substantial interest from industry and public health authorities, and there are a number of studies focused on the development of nanomaterials using more economic methods and sources. Nanosensors allow for rapid in-field detection and real-time monitoring that simply do not seem possible using conventional analytical approaches. This convergence of nanosensor technologies and plant sciences could support the successful delivery of major public goals such as the 2030 Sustainable Development Goals. It is envisaged that the superior performance afforded by nanosensors will be combined with smart technology and the Internet of Things (IoT) to meet demand gaps, help increase production, and generate valuable data. Advances, such as energy harvesting and the application of technologies such as fuel cells [229] to meet the power requirements, will also become more important.

## 4. Summary

Plant science has a role in the production of staple foods and materials, as well as roles in genetics research, environmental management, and the synthesis of high-value compounds. Nanosensors can help to address some of the most significant challenges we currently face, such as energy and food security, by providing insights that can be exploited to support plant growth. The assessment of plant characteristics is vital to determine whether plant breeding programmes have resulted in the incorporation of desirable traits in plants. The application of nanosensors in plant science offers opportunities to study the distribution and transport of various analytes in vivo, as well as plant signalling, and plant responses to environmental conditions. In the field, nanosensors could be used for nutrient analysis to determine if supplementation is required for optimal plant growth, and they offer the opportunity to detect pathogens so that containment is possible.

## Figures and Tables

**Figure 1 biosensors-12-00675-f001:**
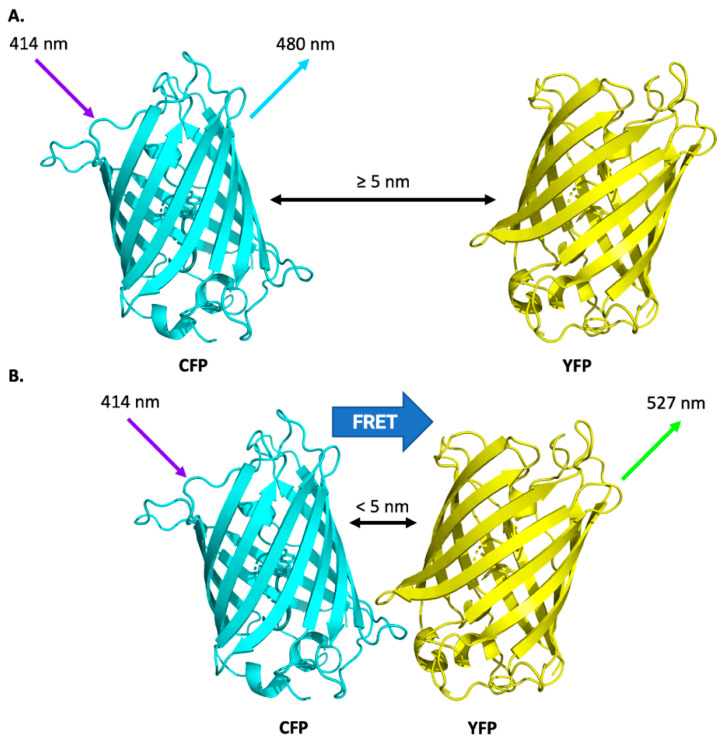
FRET using the CFP/YFP donor/acceptor pairing. (**A**) Excitation of the donor molecule (CFP using light with a wavelength of 414 nm) only produces observable emission (at 480 nm) from the donor if the two fluorescent moieties are too far apart. (**B**) Excitation of the donor is propagated to the acceptor molecule (YFP) via non radiative dipole–dipole coupling when within range and emission (at 527 nm) from the acceptor is observed.

**Figure 2 biosensors-12-00675-f002:**
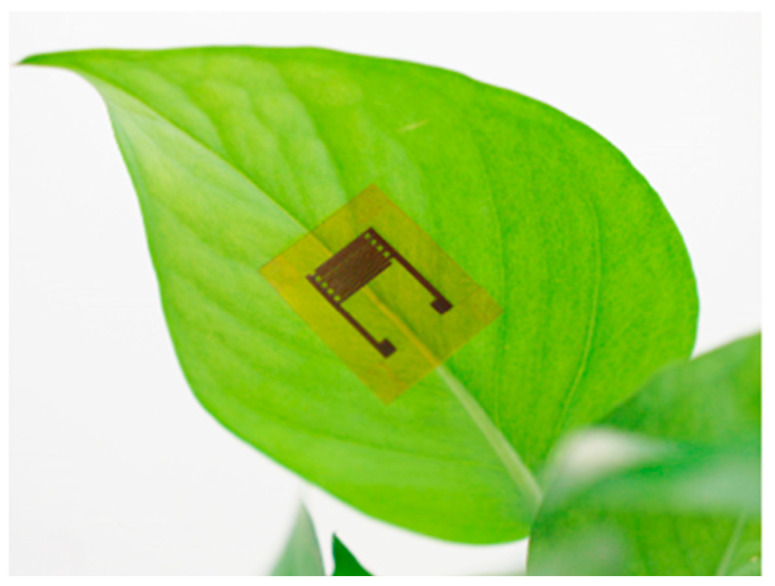
A photograph of the graphene oxide-based humidity sensor attached to the lower surface of a leaf (reproduced from Lan et al. 2020 [157] with permission).

**Figure 3 biosensors-12-00675-f003:**
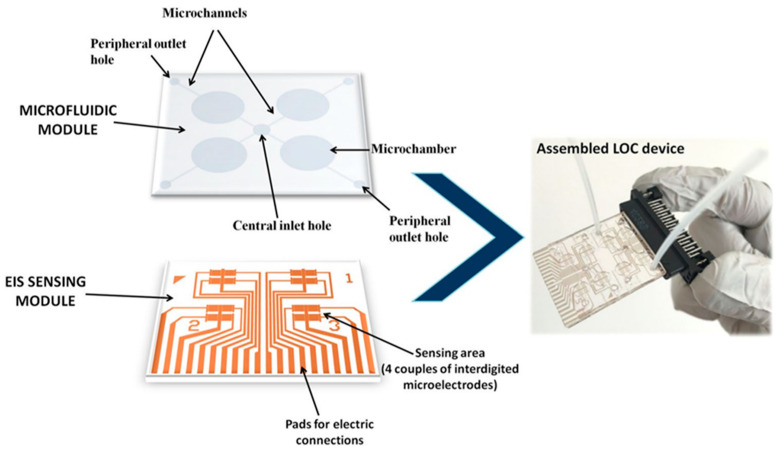
Description of the lab-on-a-chip device for the detection of *Xylella fastidiosa* made up of a sensing and a microfluidic module [217] with permissions.

**Table 1 biosensors-12-00675-t001:** Nanosensors, their mechanism of action, and example analytes in plants.

Sensor Type/Detector	Mechanism	Analytes in Plants
Förster Resonance Energy Transfer (FRET)	A recognition element is fused to a reporter element (this is a fluorophore pair that have an overlapping emission spectra). The donor chromophore in its excited state may transfer energy to an acceptor chromophore through nonradiative dipole–dipole coupling.	ATP, calcium ions, metabolites, transgenes, and plant viruses.
Surface-Enhanced Ramen Scattering (SERS)	A technique that enhances Raman scattering by molecules adsorbed on rough metal surfaces or by nanostructures. The enhancement factor can be as much as 10^14^, and hence the technique may detect single molecules.	Hormones, e.g., cytokinins and brassinosteroids, as well as pesticides.
Electrochemical	Comprises a working electrode, counter electrode, and reference electrode. Reports the electrochemical response or electrical resistance change of materials resulting from a reaction with the analytes.	Hormones, enzymes, metabolites, ROS, and ions such as H^+^, K^+^, and Na^+^.
Piezoelectric	A reversible process in which mechanical stress is converted into an electric signal.	Morphogenesis.

**Table 2 biosensors-12-00675-t002:** Förster resonance energy transfer-based nanosensors and plant analytes.

Förster Resonance Energy Transfer-Based Nanosensors				
Plant Analyte	Sensor	Type	Plant Species	References
Nucleic acid	GFP-tagged proteins	Genetically encoded	*Nicotiana benthamiana*	Camborde et al., 2017 [41]
Glucose	FLIP: FRET between a cyan fluorescent protein and a yellow fluorescent protein	Genetically encoded	*A. thaliana* and *Oryza sativa* L. spp. *japonica* cv. Zhonghua11	Chaudhuri et al., 2011 [42] and Zhu et al., 2017 [43]
ATP	Nano-lantern: a chimera of enhanced Renilla luciferase and the fluorescent protein Venus	Genetically encoded	*A. thaliana*	Saito et al., 2012 [44]
Ca^2+^ ions	Yellow cameleons: FRET between a cyan fluorescent protein and a yellow fluorescent protein	Genetically encoded	*Lotus japonicus*	Krebs et al., 2012 [45]
Plant hormone: Gibberellin	FRET between a cyan fluorescent protein and a yellow fluorescent protein	Genetically encoded	*A. thaliana*	Rizza et al., 2017 [46]
Plant virus: Citrus tristeza virus	Carbon nanoparticles acting as quenchers and antibodies labeled with CdTe quantum dots	Exogenously applied	*Citrus sp.*	Shojaei et al., 2016 [47]
Plant virus: Grapevine virus A-type	Films of zinc oxide deposited by atomic layer deposition	Exogenously applied	*Vitis sp.*	Tereshchenko et al., 2017 [48]
Transgenes/virus: Cauliflower mosaic virus 35s	DNA hybridization with probe modified nitrogen-doped graphene quantum dots and silver nanoparticles	Exogenously applied	*Glycine max*	Li et al., 2016 [49]

**Table 3 biosensors-12-00675-t003:** Surface-enhanced Raman spectroscopic analysis of plant analytes.

Surface-Enhanced Raman Spectroscopic-Based Nanosensors				
Plant Analyte	Nanomaterial	Detection Limit	Plant Species	References
Hormone: indole-3-butyric acid	Gold (Au) nanoparticles	0.002 μM	Pea, mungbean, soybean, and black bean	Wang et al., 2017 [86]
Hormone: Brassinosteroids	Au nanoparticles	1 × 10^−11^ M	Not specified	Chen et al., 2017 [87]
Pesticide: N^6^-benzylaminopurine	Au colloidal nanoparticles	0.065 μg/g	Commercial bean sprouts and bean grains	Zhang et al., 2018 [88]
Pesticide: parathion-ethyl	Plasmonic silver nanoaggregates	0.1 ppm	Apple	Li et al., 2022 [89]

**Table 4 biosensors-12-00675-t004:** Plant analytes determined with electrochemical nanosensors.

Electrochemical Nanosensors				
Plant Analyte	Nanomaterial	Detection Method	Detection Limit	References
Hormone: indole-3-acetic acid	Multi-walled carbon nanotubes	Amperometry	0.4 μM	McLamore et al., 2010 [98]
Hormone: indole-3-acetic acid	Microelectrodes decorated with nanowires	Amperometry	1 nM	Liu et al., 2014 [99]
Hormone: Ethylene	Chemoresistive sensor modified with organo–copper complex and single-walled carbon nanotubes	Chemoresistivity	<0.5 ppm	Esser et al., 2012 [122]
Hormone: Ethylene	Metal-stabilized thiyl radical film chemoresistive sensor	Chemoresistivity	30%	Chauhan et al., 2014 [123]
Enzyme: Urease	Nickel nanoelectrodes	Differential pulse voltammetry	200 ng/mL	Hubalek et al., 2007 [101]
Vitamin C	Immobilized ascorbate oxidase in poly(3,4-ethylenedioxythio-phene)-lauroylsarcosinate film electrode	Amperometry and voltammetry	Amperometry 0.464 μM; voltammetry 56.1 μM	Wen et al., 2012 [104]
Molecular oxygen	Carbon-filled quartz micropipettes with Platinum-coated tips (tip diameter in the nanometre range)	Cyclic voltammetry	-	Alova et al., 2020 [114]
Oxidation: Hydrogen peroxide	Multi-walled carbon nanotubes	Amperometry	0.27 μM	Nasirizadeh et al., 2016 [110]
Oxidation: Hydrogen peroxide	Platinum (Pt) nanoparticles	Amperometry	5.0 × 10^−9^ M	Ai et al., 2009 [111]
Antioxidant: Glutathione	Glutathione peroxidase Pt nanoparticle glassy carbon paste electrode	Differential pulse voltammetry	-	Anik et al., 2016 [117]
Ions: Cd(II), Cu(II), and Pb(II)	Multi-walled carbon nanotubes	Cyclic voltammetry	Cd(II): 1.03 μg L^−1^ Cu(II): 2.12 μg L^−1^ Pb(II): 1.62 μg L^−1^	Roy et al., 2014 [120]
Plant virus: *Pseudomonas syringae*	Gold nanoparticles	Differential pulse voltammetry	-	Lau et al., 2017 [16]

## Data Availability

Not applicable.

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
