# Peer review of "Nanosensor Applications in Plant Science"

_biosensors, 2022, doi:10.3390/bios12090675_

Round 1
Reviewer 1 Report
The authors focus on the application of nanosensors in living plants, plant cells, plant tissues, and plant organelles. After throughly reviewed the manuscript, although author present a number of sensors and applications, the Tables, Figures and Contents to be improved. Therefore, I'd recommend major revisions before possible publication. Specific comments are given below:
Abstract
Line 17: Please give the full name of FRET when it first appears.
Line 20: The last sentence seems to be missing the summary.
1. Introduction
Line 24-31: I don't quite understand why this paragraph was put here.
Lack of comparison with traditional testing methods in the introduction section.
Lack of information about other reviews in this field and how authors differ from them.
2. The designs and principles of nanosensors used in plant science
This section suggests that the author compare the advantages and disadvantages of each method (FRET, SERS, Electrochemical & Piezoelectric). For example, they are time consuming, required costly equipment, produced false negative results from cross contamination, and need professional experts.
Suggestions for drawing the basic components of a nanosensor.
Line 69: It is recommended that the contents of this table continue to be refined. Nanomaterial? Test host(s)? Type of study? Detection limit? Duration of assay? Sensitivity?
Summary
Future directions for research should also be mentioned.
Author Response
Reviewer 1
The reviewer has asked us to improve the figures and tables in our review. We have now included a further new two figures and updated the table we originally presented, but would like to us address a few points before publication. Our replies below are given in the same order as given by the reviewer.
Abstract
The reviewer has asked us to give the full name for the abbreviation FRET used the abstract, Line 17: We have now done so.
The reviewer has asked us to included summary section in the Abstract we have now done so in lines 21-24.
- Introduction
The reviewer has asked us to delete the first paragraph of the Introduction section, Lines 28-35: We have now done so.
The reviewer has asked us to include some more comparisons with traditional testing methods. We have now done so on lines, 464-466; 505-508 and 531–538.
The reviewer has commented on a lack of information about other reviews in this field and how our review differs from that given previously. We have now added a further seven references in the Introduction section detailing a number of previous reviews and highlighting how these differ from the present submission (lines 58-61).
- The designs and principles of nanosensors used in plant science
The reviewer has asked us to include some basic designs of the components of a nanosensor. We believe that this is beyond the remit of this review focused on nanosensors for plant sciences applications and their inclusion would dilute the impact of paper, as it would weaken its focus.
The review has asked to include further details to our table 1. They have asked us to include further specific details, as limit of detection, plant type, etc. we included this table to give an overview of the varied applications that have been used. The inclusions of specifics such as that asked by the reviewer would consequently, not work with our table design, or with its aim.
Summary
The review has asked us to include some ideas on the future directions for research. We have now included a further new section; 3.11 Future Directions to discuss this on page 11.
Reviewer 2 Report
Shaw et al. demonstrated the application of various nanosensors in the plants. Although authors have shown the different methods, but it need to include some more modifications as suggested below:
1. Author should include more figures describing the nanosensor in plant application.
2. Please include tables which shows different nanosensors and their application.
3. Various references cited are more than 6-year-old, please update the reference list, if possible please add some references of 2022.
4. Limitations, conclusion and future prospects of nanosensors in plant should be added.
5. In each section, author should add discussion of few recent papers related to that topic.
6. Section with colorimetric sensing, optical fiber based sensing will be useful.
7. Author should add section with advantage of nanosensor in prevention of fruit and vegetable ripening.
8. If possible, please add section with lab-on-chip example in plant sensing.
Author Response
Reviewer 2
The reviewer has stated that we have successfully demonstrated the application of various nanosensors in the plants, but would like to us address a few points before publication. Our replies below are given in the same order as given by the reviewer.
- The reviewer has asked us to include some more figures. We have now included a further two new figures and text to illustrate these on pages 7 and 10.
- Please include tables which shows different nanosensors and their application.
- The reviewer has asked for the inclusion of further references more recent than 2016. We have now included a further eleven references to meet with this.
- The reviewer has asked for some discussion on the limitations, conclusion and future prospects of nanosensors in plant sciences to be added. We have now included a new section 3.9 Future Directions to include this.
- The reviewer has asked us to include some discussion on some more recent papers related to that topic. We have now added further 13 references; 11 of these are post 2016 to address this point.
- the reviewer has asked us to include an additional topic of optical fibre-based sensing. We have now done so in lines 339-367.
- The reviewer has asked us to add section with advantage of nanosensor in prevention of fruit and vegetable ripening. We have now added a new section, 3.8 Determination of fruit ripening to address this.
- The reviewer has asked us to add material on lab-on-a-chip applications for the analysis of plants. We have now added a further four references focused on this application.
Round 2
Reviewer 2 Report
Authors have not revised the manuscript properly. I have not found the answer to comment 2 (Please include tables which shows different nanosensors and their application.) .
Author Response
The reviewer has high lighted that we have not addressed their comment 2 regarding the inclusion of tables to show different nanosensors and their application.
We are very sorry for the oversight and have now included three new tables to highlight and summarise the nanosensors described in the review paper. These are given as table 2 (page 3, lines 103); table 3 (page 6, lines 195) and table 4 (page 7, lines 237) in the updated submission.
We have also proof read again the paper and corrected a few minor grammatical/typos.